# Green Oxidative Catalytic Processes for the Preparation of APIs and Precursors

**Pedro D. García-Fernández, Juan M. Coto-Cid and Gonzalo de Gonzalo ***

Department of Organic Chemistry, University of Seville, c/Profesor García González 1, 41012 Seville, Spain
* Correspondence: gdegonzalo@us.es; Tel.: +34-955-420-802

**Abstract:** Asymmetric oxidation processes have constituted a valuable tool for the synthesis of active pharmaceutical ingredients (APIs), especially for the preparation of optically active sulfoxides, compounds with interesting biological properties. Classical approaches for these oxidative procedures usually require the application of non-sustainable conditions that employ hazardous reagents and solvents. In the last decades, chemists have tried to combine the preparation of valuable compounds of high yields and selectivities with the development of more sustainable protocols. To achieve this objective, greener solvents, reagents, and catalysts are employed, together with the use of novel chemical techniques such as flow catalysis or photocatalysis. The last efforts in the development of greener approaches for the preparation of APIs and their intermediates using oxidative procedure will be reviewed herein. Most of these approaches refer to biocatalytic methods, in which mild reaction conditions and reagents are employed, but other novel techniques such as photocatalysis will be described.

**Keywords:** active pharmaceutical ingredients; oxidations; biocatalysis; Green Chemistry; photocatalysis; biobased solvents





## 1. Introduction

Active pharmaceutical ingredients (APIs) can be defined as those biological compounds used for the detection, prevention, and treatment of different types of diseases [1]. Due to the current diseases and the continuous emerging ones, the API production sector is continually growing within the global market, comprising almost 190 million dollars in 2020 and assuming an increase of 6.6% expected per year [2]. Factors such as high production costs and environmental sustainability are taken with more awareness and the recent COVID-19 pandemic have produced a change in the manufacture of APIs, promoting their importation from foreign countries. To return to local manufacturing, numerous state agencies are developing various initiatives that promote the production of APIs by following the principles of Green Chemistry with the aim of not harming the environment and complying with environmental regulations.

In the final years of the last century, it became clear that chemistry required an evolution, adapting the classical concepts of efficiency and selectivity to more sustainable procedures in which novel parameters were considered, including the use of raw starting materials, the application of more sustainable and non-hazard compounds, and the reduction of the raw disposals [3–5]. Thus, in early 90's, the Green Chemistry concept was developed, but it was not until 1998 that Anastas and Warner established the Twelve Principles of this concept (Scheme 1) at the United Estates Environmental Protection Agency (US-EPA) [6,7]. In 2015, the United Nation members adopted the seventeen United Nation Sustainable Development Goals (UN SDG) [8], whose objective number 3 ensured the healthy life and the promotion of wellness for all ages. Likewise, objective number 12, which ensured sustainable consumption and productions patterns, led to industrial chemical and pharmaceutical companies to look for more sustainable procedures. These objectives, in addition to the Green Chemistry's principles, provide a reflection on the

pharmaceutical production conditions [9–11]. Taking as reference the Green Chemistry Principles, laboratory and industrial habits can begin to be modified for the development of novel reagents, catalysts, and solvents. The solvent in which chemical transformations are performed is of fundamental importance [12,13]. One of the principles of Green Chemistry states that "solvents should be avoided whenever possible". However, chemical reactions generally require the presence of solvents, not only as a medium used to perform the reactions but also to develop the separation and/or purification of the reaction products, representing more or less 80% of the material used in a reaction. The most typical green solvent is obviously water, but its use presents some drawbacks in organic synthesis due to the low solubility in water of several organic molecules and the intrinsic reactivity that water can present with some functional groups. Therefore, novel solvents such as ionic liquids (ILs) and eutectic liquids (DESs) have been developed [14,15]. In addition, the use of so-called biobased solvents, that is, those organic solvents obtained from renewable sources, present interest in the field of chemical synthesis [16,17]. These biobased solvents show valuable physicochemical properties that result in a low environmental impact, so their use instead of classic organic solvents is of great interest for the implementation of green procedures.

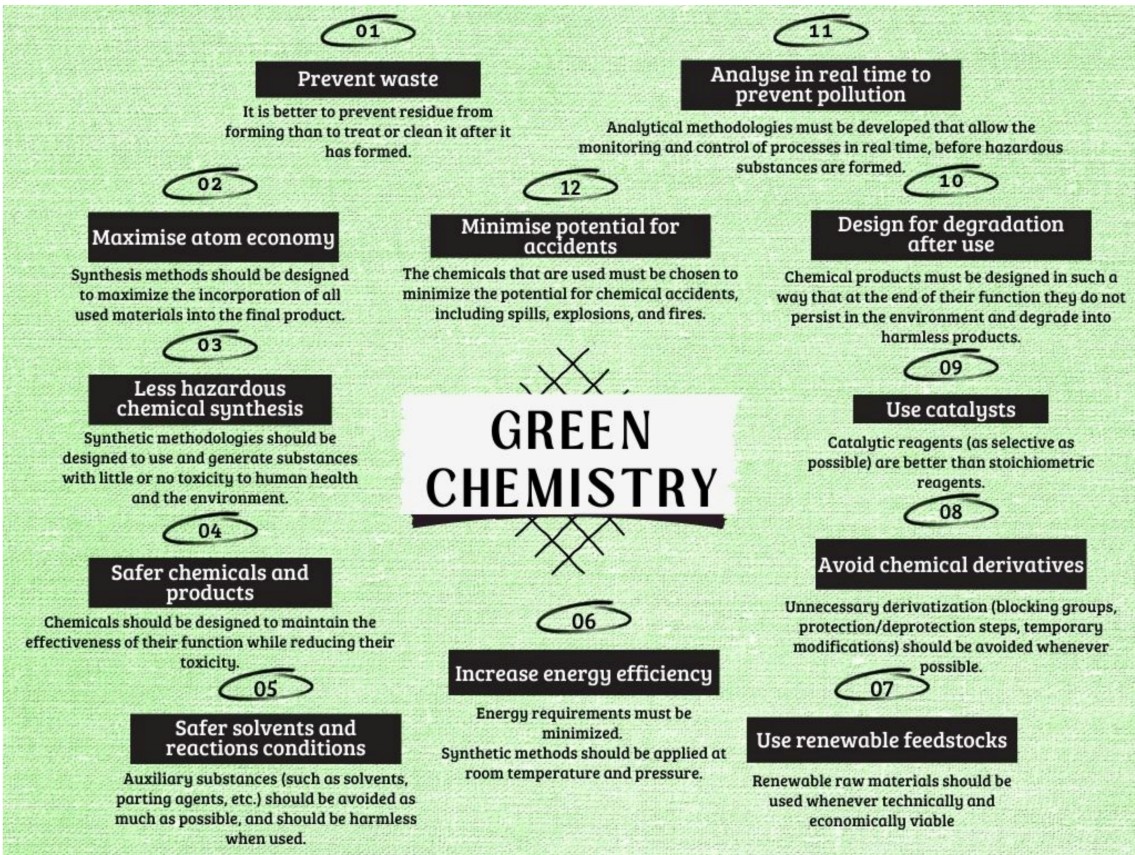

**Scheme 1.** The Twelve Principles of Green Chemistry.

Another aspect in which chemists can develop greener procedures consists in the application of more efficient chemical syntheses, for instance, by presenting a lower number of steps than the classical procedures. The preparation of the well-known drug Ibuprofen, commonly used as a pain killer, is a good example. This API was first patented by the Boots company in the 1960s in a six-step procedure as shown in Scheme 2 [18].

**Scheme 2.** Established protocols for the preparation of ibuprofen by (**a**) the Classical six-step procedure and (**b**) the Greener three-step synthesis including a palladium-catalyzed oxidation.

After the patent ran out, the Boots-Hoechst-Celanese (BHC) company developed a novel synthetic methodology for producing ibuprofen from the same starting material and remarkably reduced the number of steps from six to three [19]. The last step consists of a palladium-catalyzed oxidation performed in water as the "green" solvent that affords the desired compound [20]. This latter protocol led to a more sustainable procedure by reducing the amount of waste and the number of purification steps in the process. In terms of atom economy [21], in the classical approach, the overall atoms used were around 40%, while the three-catalytic-step green procedure results in 77% atom utilization, representing a great improvement and a more sustainable procedure for the synthesis of this valuable API.

Most of the different oxidative catalytic procedures described herein are devoted to the preparation of optically active sulfoxides. These compounds, apart from being employed as valuable chiral ligands and auxiliaries in asymmetric synthesis, are present in several molecules with pharmacological activity [22–25]. Thus, some sulfoxide-containing molecules such as prazoles (omeprazole, lansoprazole . . . ), modafinil, sulforaphane, sulindac, and others have been widely employed as valuable drugs. The typical methodology employed for the preparation of asymmetric sulfoxides rely on the selective oxidation of prochiral sulfides. For this reason, the search for sustainable sulfoxidation procedures is one of the targets of this revision; this also shows the latest achievements for the preparation of Active Pharmaceutical Ingredients employing catalytic oxidative procedures in sustainable conditions and accomplished through the Green Chemistry principles.

## 2. Biocatalytic Approaches for the Oxidative Preparation of APIs and Precursors

There are different types of catalysts employed in chemical synthesis, but enzymes in different preparations (well as free biocatalysts, cells free extracts, or whole cells systems) have attracted great interest since the last years of the past century [26–29]. In the mid–1980s, the development of cloning and overexpression techniques, the advances in immobilization procedures, and the application of enzymes in non-aqueous systems have allowed the development of several enzymatic methodologies for the synthesis of high added-value compounds, including the preparation of APIs [30–34]. Biocatalyzed reactions usually occur with high efficiency, excellent selectivity, good yields, environmental sustainability, and lower costs, which make them more attractive from an industrial perspective. However,

it must be taken into account that these procedures also present some drawbacks, such as the (relatively) high substrate specificity of biocatalysts or the low substrate loadings required due to their generally low solubility in water.

Oxidations carried out by biological systems present several advantages regarding classical methodologies [35,36]. Thus, biodegradable, non-toxic, and non-hazard catalysts are employed, whereas mild reaction conditions and molecular oxygen or hydrogen peroxide as mild oxidants are used. Oxidoreductases (E.C. 1.x.x.x) can perform the biocatalytic oxidation of different functional groups, including alcohols, amines, carbonyl compounds, and heteroatoms. This class of enzymes comprises different groups of biocatalysts, including dehydrogenases, oxidases, peroxidases, and oxygenases (both mono- and di-), presenting different properties and catalytic activities.

### 2.1. Oxidations Catalyzed by Monooxygenases

Monooxygenases can catalyze the insertion of one atom from molecular oxygen into different molecules, whereas the other oxygen atom is released as water [37]. Among these enzymes, Baeyer-Villiger monooxygenases (BVMOs) are monooxygenases that contain flavin as prosthetic group and are able to catalyze the Baeyer-Villiger reaction and boron atom oxidation, as well as the oxygenation of different heteroatoms (sulfur, nitrogen, or phosphorous). BVMOs require nicotinamides (generally NADPH) as electron source for carrying out their activity. Due to its high price and instability, this cofactor must be recycled through a secondary enzymatic system as formate dehydrogenase (FDH), glucose or glucose-6-phosphate dehydrogenase (GDH or G6PDH), phosphite dehydrogenase (PTDH), or a ketoreductase (KRED). BVMOs have been employed for the preparation of optically active sulfoxides from prochiral sulfides in oxidative processes under mild and environmentally friendly conditions [38–40].

One example has been recently shown in the BVMO-catalyzed sulfoxidation step in the synthesis of AZD6738 (**3**), a candidate drug for the treatment of colon and hematological cancers [41]. This drug presents a chiral sulfoximine moiety in its structure, which is obtained from sulfoxide (*R*,*R*)–**2**. The initial synthesis of **2** from its prochiral sulfide **1** is carried out by employing *m*-chloroperbenzoic acid, which requires a chromatographic separation of the diastereomers obtained, yielding 50% of the desired sulfoxide as the highest yield. The application of the Kagan procedure in the presence of a titanium catalyst, afforded **2** in low yields, with degradation products. For these reasons, the biocatalytic approach for the oxidation of **1** was applied [42]. Reaction optimization was carried out using the design of experiments (DOE), analyzing the effect of substrate and enzyme concentrations, cosolvent charge, the type of buffer, and enzyme type. After analyzing more than 100 BVMOs, the best results were obtained with BVMO P1/D08 from the company Codexis in triethanolamine (TEA) buffer at pH 9.0, obtaining a 94% conversion starting from a sulfide concentration of 40 g/L and a BVMO loading of 24% weight (Scheme 3). The reaction was scaled up to the kilogram scale, analyzing all the parameters that could affect the process. Thus, an efficient agitation (500 rpm) and gas-liquid mass transfer (stream of air to supply $O_2$ to the reaction) were required, as well as successive additions of *iso*-propanol (IPA), which was employed as cosolvent and cosubstrate to regenerate the cofactor NADPH in combination with the KRED CDX-019 to offset evaporation. Enzyme loading could be optimized up to 5% weight, achieving a 77% isolated yield after a one-day reaction. A further development in the biocatalyzed process was performed by carrying out the oxidations at higher scale (60 kg batches), leading to a 74% yield of the chiral sulfoxide with >99% enantiomeric excess (*ee*), which improves to a great extent the precedent chemical-synthesis protocols and increases the sustainability of the process.

**Scheme 3.** Biocatalyzed synthesis of sulfoxide (*R,R*)–**2** employing a BVMO.

Esomeprazole, the most valuable compound from the family of prazoles, is the (*S*) enantiomer of omeprazole (**5**), the first drug of the class of proton-pump inhibitors (PPIs), which mechanistically inhibit gastric-acid secretion and are thus used as antiulcer agents [43,44]. This drug is commercialized as Nexium®, as the key step in its preparation is the selective oxidation of the prochiral sulfide pyrmetazole (thioether-5-methoxy-2-[((4-methoxy-3,5-dimethyl-2-pyridinyl)methyl)thio]–1*H*–benzimidzole, **4**). The classical methodology for this oxidation required a modification of the Kagan method [45], employing a titanium-catalyzed procedure in the presence of cumene hydroperoxide. This methodology presents some drawbacks, such as the formation of the sulfone overoxidation product or the use of harsh reagents and conditions. For these reasons, a biocatalytic approach for the preparation of esomeprazole from pyrmetazole at a high scale was developed in 2018 by Codexis Inc. [46], employing the cyclohexane monooxygenase from *Acinetobacter calcoaceticus* NCIMB 9871 (CHMO), the most versatile BVMO, capable of catalyzing the oxidation of a wide set of ketones, sulfides, and other compounds [47].

Wild-type CHMO was able to catalyze the pyrmetazole oxidation with very low conversion. Directed evolution was performed on this biocatalyst to obtain an improved CHMO mutant with higher enzymatic productivity and an excellent enantioselectivity with no sulfone overoxidation-product formation. Thus, several thousands of variants were produced over 19 rounds of evolution. After combining the beneficial mutations in the subsequent rounds, it was possible to achieve a CHMO variant that improved the wild-type enzyme 140,000 times over. After finding the best biocatalyst for this process, other critical oxidation parameters, including the oxygen supply, substrate concentration, the addition of catalase to remove the hydrogen peroxide formed in the unspecific oxidation, and the presence of IPA and a KRED for the cofactor regeneration, were analyzed to find a suitable procedure for the preparation of (*S*)–**5**. Thus, the reaction was carried out in a 30-g scale (50 g/L pyrmetazole) in buffer phosphate of pH 9.0 containing 4% *v*/*v* IPA, the BVMO, KRED CDX-019, catalase, NADP$^+$, and esomeprazole seed. The final product was recovered after extraction using isobutyl methyl ketone with 99% purity in 87% isolated yield and 99.9% *ee*. This result represents a good alternative to the Kagan procedure in terms of productivity, cost, and environmental impact, but it is still insufficient for developing an effective high-scale process.

One year later, a set of mutants from a CHMO from *Acinetobacter calcoaceticus* (a BVMO with 70% identity regarding the NCIMB 9871 variant) were prepared by modifying the biocatalyst's active sites close to the substrate tunnel. The use of a structure-based protein-engineering approach made possible the preparation of a CHMO mutant (CHMO$_{M6}$) that presented a 5611-fold increased activity on omeprazole respecting the starting biocatalyst (Scheme 4) [48]. After 22 h, this variant was able to convert a 96% or starting **4** (3.0 g/L), making it a promising candidate for applications at an industrial scale.

**Scheme 4.** General method for the biocatalytic synthesis of (*S*)-omeprazole employing BVMOs.

This novel variant, also called prazole sulfide monooxygenase (*Ac*PSMO), was later employed in the (*S*)-omeprazole synthesis at a 300 L scale. The reaction was carried out in a buffer of pH 8.0 containing a 5% *v/v* methanol to ensure a proper substrate solubilization in the reaction medium. NADPH-cofactor regeneration was carried out by employing sodium formate and FDH, as the formate oxidation afforded $CO_2$ in an irreversible process. Since the oxygen supply is another critical factor of the process, pressurized air was employed to increase the volumetric transfer coefficient and the saturation concentration. The biooxidation was carried out starting from 600 g of pyrmetazole, and once completed, NaOH was added to the reaction to yield the omeprazole sodium salt, which was removed from the crude mixture by membrane filtration, resulting in a great reduction of the ethyl acetate employed in the product downstream. After drying, 380 g of esomeprazole (58% yield) were recovered with a 99.1% purity and more than 99% enantiomeric excess [49]. This procedure represents the first green-by-design system for the large-scale production of esomeprazole and other PPIs, but it still requires optimization, especially regarding the biocatalyst activity.

(*R*)-Lansoprazole (**7**) is also a valuable PPI, but the different attempts to obtain this enantiomerically pure API through biocatalytic methods were unsuccessful. In 2018, two Baeyer-Villiger monooxygenases, one from *Bradyrhizobium oligotrophicum* (*Bo*BVMO) and the other from *Aeromicrobium marinum* (AmBVMO), were employed to catalyze its formation starting from the lansoprazole sulfide with high selectivity [50]. These biocatalysts were also able to catalyze the formation of other prazoles, including omeprazole, rabeprazole, pantoprazole, and ibaprazole. A further improvement in the biocatalytic preparation of (*R*)-lansoprazole from its prochiral sulfide **6** was established in 2022 [51]. After analyzing different biocatalysts, the BVMO form *Cupriavidus basilensis* (*Cb*BVMO) showed the highest activity and selectivity in this oxidative procedure. This BVMO can convert 10 mM of the starting material to (*R*)-lansoprazole completely and with 99% *ee* after 35 h of employing the sodium formate/FDH system for the regeneration of the NADPH cofactor (Scheme 5). The excellent selectivity featured by the catalyst and complete conversion in the reaction were also observed for the formation of other prazoles, including (*R*)-omeprazole, (*R*)-pantoprazole, and (*R*)-raboprazole, but the biocatalyst is still being improved in order to achieve higher catalytic efficiencies that will allow the application of this process at an industrial level.

**Scheme 5.** Synthesis of (*R*)-lansoprazole catalyzed by *Cb*BVMO.

Apart from those monooxygenases containing flavin as a prosthetic group, heme-dependent monooxygenases are also valuable biocatalysts, catalyzing a wide set of oxidative processes. Cytochrome P450 (CYPs) enzymes belong to this family and are

employed as versatile enzymes for the regio- and/or stereoselective functionalization of non-activated carbon atoms while using mild and environmentally friendly reaction conditions [52]. The main problem for these enzymes is that they usually present low activity and thermostability, which makes difficult its application at the industrial scale, making it necessary to prepare improved mutants with enhanced properties [53]. Some of these biocatalysts have been employed for the selective hydroxylation of steroids, compounds with high biological activity widely employed in pharmaceutical industry. CYP102A1, obtained from *Bacillus megaterium* (P450$_{BM3}$), is considered one of the most active CYPs [54], but the wild-type catalyst is not active on steroids. In 2018, two P450$_{BM3}$ mutants, W1F1-WC and WWV-QRS, were found to selectively oxidize testosterone (**8**) at the C16 position, recovering the final products 16α- (**9**) and 16β-hydroxytestosterone (**10**) with a selectivity higher than 92% (Scheme 6a) [55]. CYP106A2, another P450 also found in *Bacillus megaterium* [56], was able to catalyze the oxidation of progesterone (**11**) to 6β-hydroprogesterone (**12**), but with a low selectivity (Scheme 6b). Mutagenesis studies found that the A243S mutant was able to perform this oxidation with a high selectivity, up to 86% [57]. Finally, CYP260B1 from *Soranghium cellulosum* catalyzed the hydroxylation of cortodoxone (**13**) with 95% conversion [58]; 6β-hydroxycortodoxone was the main product with 94% selectivity (Scheme 6c). The T224A mutant showed an improved selectivity in the hydroxylation of 11-deoxycorticosterone, recovering 75% of 9α-hydroxy-11-deoxycoricosterone (**14**) [59].

**Scheme 6.** CYP-catalyzed hydroxylations of steroids. (**a**) Selective oxidation of testosterone (**8**); (**b**) Biooxidation of progesterone (**11**), and (**c**) Biocatalyzed hydroxylation of cortodoxone (**13**).

## 2.2. Other Biocatalyzed Oxidations

Islatravir (**17**) is a nucleoside analogue with high anti-HIV activity [60]. Several synthetic routes have been proposed for its preparation, but most of them require several steps with multiple protection/deprotection procedures. The biocatalytic approach proposed for the synthesis of **17** reduced the number of steps, making it possible to achieve the final product using a nine-enzyme system when starting from ethynyl glycerol **15** (Scheme 7) [61]. The initial step of this process consists in the selective oxidation of **15** to an aldehyde **16** catalyzed by a galactose oxidase from *Fusarium graminearum*. Oxidases are a class of oxidoreductases able to catalyze different oxidative reactions by employing molecular oxygen at mild reaction conditions [62–64]. The substrate oxidation and the reduction of molecular

oxygen occur in the same active site, forming hydrogen peroxide as a byproduct and not requiring any cofactor for their activity. These enzymes have been employed in different types of oxidative procedures. Initial experiments were carried out with a galactose oxidase variant, which catalyzed the formation of the undesired (*S*)-enantiomer of compound **16** with moderate conversion, so several rounds of evolution were required to obtain mutants with improved activity, lower product inhibition, and a reversal on the enantioselectivity by mutating the positions W290 and F464. Oxidations were carried out in the presence of two additional enzymes: catalase, which was used to eliminate the hydrogen peroxide formed in the medium, and peroxidase, used to maintain the proper copper-oxidation state. Once the reaction conditions were optimized, it was possible to achieve (*R*)-**16** with 97% enantiomeric excess and a high conversion.

**Scheme 7.** Galactose oxidase enantioselective oxidation of islatravir precursor **15**.

Furthermore, the manufacture of APIs on an industrial scale has been recently improved by employing continuous flow synthesis. This allows a larger scale production with better atomic efficiency, in addition to facilitating the automation of the synthetic process [65–68]. One example has been recently shown in the preparation of captopril (**20**), an angiotensin-converting enzyme inhibitor employed for the treatment of hypertension [69]. This API was discovered and developed at E. R. Squibb & Sons Pharmaceuticals in the 1970s and is still prescribed [70]. In 2021, Romano et al. designed a synthetic route under continuous flow for captopril synthesis in which the first reaction was a biocatalytic oxidation followed by three chemical transformations until the enantiomerically pure active ingredient was obtained, as shown in Scheme 8 [71]. The oxidation of commercially available diol **18** was carried out using a continuous flow of 1.0 g/L of starting diol in an acetate buffer of pH 6.0 catalyzed by *Acetobacter aceti* MIM 2000/28, an acetic acid bacterium. These types of bacteria can perform the chemo-, regio-, and stereo-selective oxidation of primary alcohols, affording the corresponding carboxylic acids with good yields, while aldehydes were not normally isolated [72]. The biocatalyst was immobilized by entrapment in dry cells of alginate. After this, it was packed into a glass column with an inner diameter of 15 mm and swelled by passing buffer through the column. The air supply was maintained using an air-liquid flow stream. After 10 min, carboxylic acid **19** was recovered with 95% conversion and 97% enantiomeric excess. By employing 400 mg of the immobilized enzyme, around 50 mL was collected, achieving the highest conversion rate in the first 35 mL. In order to recover the final product, a catch-and-release system was employed using an Ambersep 900 OH resin-packed column in which the acid was entrapped, and afterwards, eluted by treatment with HCl 1.0 N.

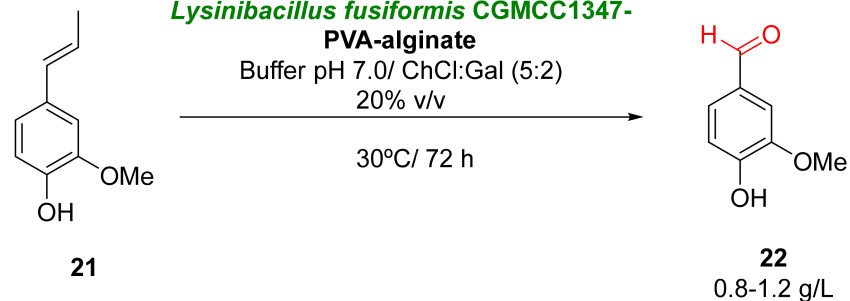

**Scheme 8.** Flow synthesis of carboxylic acid precursor of captopril by enzymatic oxidation catalyzed by *Acetobacter aceti* MIM 2000/8.

Vanillin (4-hydroxy-3-methoxybenzaldehyde) is one of the most used flavors in the world, but apart from this use, it has been also employed in the pharmaceutical and medical industries as it displays antioxidant, neuroprotective, and anticancer activities [73]. In 2017, different Deep Eutectic Solvents (DESs) were employed as cosolvents (1% *v/v*) in the conversion of isoeugenol (**21**) into vanillin (**22**) in a process catalyzed by *Lysinibacillus fusiformis* CGMCC1347 cells [74]. DESs are liquids prepared by mixing a hydrogen-bond donor with a hydrogen-bond acceptor (generally a quaternary amine) at a precise molar ratio at about 80 °C. The liquid formed maintains its state, presenting valuable properties as (co)solvents in catalysis. These compounds are prepared with a complete atom economy, being biodegradable, cheap, and having high thermal stability [75,76].

For almost all of the cosolvents tested, higher conversions were achieved than when employing only an aqueous buffer. Authors claimed that this improvement was due to higher cell permeability due to the DESs effect, which facilitates the entrance of the hydrophobic substrate. When the oxidations were performed at a higher DES content (20% *v/v*), higher yields were achieved with several of these compounds than in only buffer. The use of *Lysinibacillus fusiformis* cells immobilized on poly(vinylalcohol)-alginate allowed the recycling of the biocatalyst 13 times within 72 h at 30 °C in buffer of pH 7.0 containing 20% *v/v* choline chloride:galactose as DES, while maintaining the enzymatic activity (Scheme 9).

**Scheme 9.** Synthesis of vanillin (**22**) by oxidation catalyzed by employing immobilized cells of *Lysinbacillys fusiformis* in the presence of Deep Eutectic Solvents.

Armodafinil is a potent drug used as a treatment for fatigue in narcolepsy, sleep apnea, and sleep disorders [77,78]. It is the isomer (R) of the racemic compound, modafinil (**24**), which has the longest duration of both enantiomers due to a slower metabolism and excretion, resulting in a longer time of action. The group, Jiang et al., reported an enantioselective synthesis of (R)-modafinil using a chloroperoxidase (CPO) in an oxidative process (Scheme 10) [79]. These heme-dependent enzymes are capable of catalyzing halogenation reactions of organic compounds by generating reactive hypohalites in the presence of halides and hydrogen peroxide; while in the absence of the corresponding halides, CPOs are able to participate in several oxidative procedures, including hydroxylations, epoxidations, or sulfoxidations [80]. Thus, the sulfoxidation of 2-(diphenylmethylthio)acetamide (**23**) using CPO from *Caldariomyces fumago* as a oxidative catalyst and *tert*-butyl hydrogen peroxide (TBPH) as oxidizing agent in an aqueous buffer of pH 5.5 afforded (R)-modafinil with an 11% yield and 97% *ee*. This low yield is due to the low solubility of the compound **23** in an aqueous medium. For this reason, ionic liquids (ILs), quaternary ammonium salts (QAS), and polyhydroxylated compounds were employed as additives to increase the substrate solubility [81]. All of these compounds had a positive effect on the biocatalyzed system, making it possible to achieve a 41% (R)–**24** when employing the ionic liquid, 1-ethyl-3-methylimidazolium bromide ([emim]Br), while maintaining the sulfoxide's optical purity.

**Scheme 10.** CPO-catalyzed synthesis of (R)-modafinil by asymmetric sulfoxidation.

### 3. Other Sustainable Catalytic Asymmetric Oxidations

Apart from employing biocatalysts in the oxidation processes for the preparation of APIs, other approaches have been developed in recent times to get more sustainable oxidative procedures. Thus, in 2019, environmentally friendly modifications were described for the asymmetric oxidation of pyrmetazole to esomeprazole using a modification of the Kagan methodology [82]. Authors have proposed different sustainable modifications to the standard esomeprazole synthesis, including the use of water or biobased solvents, such as propylene carbonate or 2–MeTHF [83]. The use of these approaches afforded the desired compound with moderate results (up to 70% yield and 70% *ee*) but were insufficient for its application at industrial scale.

When using chemical oxidations, the application of molecular oxygen as a green mild oxidant in chemical oxidative procedures is considered a sustainable procedure, as it replaces the use of harsh oxidants, such as peroxides or peracids. The abundance and easy availability of this gas makes $O_2$ an interesting option for environmentally friendly protocols. In 2020, the aerobic oxidation was described for several prochiral sulfides and was mediated using a proline diketopiperazine dipeptide as an organocatalyst and using 1,1,1,3,3,3-hexafluoropropan-2-ol (HFIP) as the solvent [84]. The authors proposed this methodology as a green oxidative procedure for the development of further valuable sulfoxidations, but the use of hazardous solvents significantly increased the manufacturing cost.

In a recent paper, the selective and sustainable oxidation was described for the conversion of alcohols to the carbonyl compounds catalyzed by $CuCl_2$/TEMPO/TMEDA (N,N′,N′,N′-tetramethylethylenediamine) using water or the environmentally friendly low-

melting mixture (LMM) D-fructose-urea. The resulting ketones can be further converted into secondary alcohols or nitroalkanes, establishing a promising starting point for the development of valuable oxidations used to obtain APIs or their precursors [85].

Flavins have been widely employed as organic catalysts in different types of organic reactions. Due to the molecular oxygen activation of flavins, these compounds have taken part in oxidative processes, including sulfoxidation protocols [86–88]. In 2018, the application of a flavin catalyst combined with iodine for the sulfenylation reaction of imidazo[1,2–*a*]pyridines (**25**) with aromatic and aliphatic thiols to yield the sulfur compounds was described **26**, as shown in Scheme 11 [89]. The imidazo[1,2–*a*]pyridine structure is present in several valuable heterocyclic compounds with pharmaceutical activity, including olprinone (a cardiotonic compound), the anxiolytic saripidem (**27**), and antibiotics such as rifaximin or inhibitors of human rhinovirus (**28**) [90,91]. The use of a cationic flavin catalyst at 2 mol% loading in the presence of 4 mol% of iodine and oxygen, or even air, as mild oxidants in acetonitrile, afforded the sulfenylation products **26** at the 3-position with moderate to high yields (68–95%) in shorter reaction times in this metal-free procedure.

**Scheme 11.** Catalyzed oxidative sulfonylation of imidazo[1,2–*a*]pyridines (**25**) in the presence of a flavin catalyst combined with catalytic iodine.

The use of metal-based catalysts cannot be considered a completely green approach in chemical synthesis, but in the last years, some examples have appeared in which these reagents were employed under milder reaction procedures than previously described methodologies. For example, in 2013, the preparation of different chiral sulfoxides from the starting sulfides was catalyzed using a manganese compound in the presence of a phorphyrin ligand, employing hydrogen peroxide as mild oxidant [92]. When this method was tested in the oxidation of pyrmetazole (0.4 M), the use of $Mn(OTf)_2$ at 1 mol% in the presence of 1 mol% of phorphyrin and 1 equivalent of $H_2O_2$ afforded esomeoprazole with an 82% yield and 90% enantiomeric excess after 1 h. Esomeprazole can also be obtained by the oxidative kinetic oxidation of racemic omeprazole in the presence of the catalytic system. At these conditions, the desired enantiomer of the sulfoxide was recovered with 39% yield and 89% *ee*.



Recently, the use of light in chemistry has emerged as a very interesting tool used to execute different types of processes. This technique describes the chemical reactions caused by the absorption of ultraviolet (100–400 nm), visible light (400–750 nm), or infrared radiation (750–2500 nm). Despite the high energy demand of most artificial light sources, the cleanness of light and effectiveness of this process and the development of new technologies have aroused the interest of various research groups in both academia and industry in the development of sustainable procedures employing photocatalysis [93–96].

In 2022, Skolia et al. described a mild and sustainable methodology for the photochemical oxidation of sulfides to sulfoxides in the presence of molecular oxygen [97]. This methodology was further improved by combining the photochemistry approach for the light-mediated oxidation of sulfides with the use of a biobased solvent such as 2–MeTHF. This novel method was further applied to the preparation of two sulfoxides with pharmaceutical properties (Scheme 12) [98]. The first of these was sulforaphane (**30**), a compound which exhibits anticancer properties and demonstrated potential as an inhibitor of SARS-CoV-2 following in vitro replication [99]. Authors consider that they have achieved more industry-friendly results by employing LED 427 nm and anthraquinone at low loading (0.05 mol%) as a catalyst in the photooxidation of sulfide **29**. After 5 h, an 87% yield of the desired sulfoxide was obtained. Another remarkable outcome is the synthesis of armodafinil (**24**). In this case, starting sulfide **23** presented an active benzylic position in its structure, which complicates the chemoselective photochemical-oxidation protocol when applying LED 427 nm. In fact, the use of an anthraquinone-mediated photochemical protocol, even at very low catalyst loadings, leads to the decomposition of the intermediate, which is attributed to a C–S bond fragmentation in the highly reactive benzylic position. Fortunately, authors avoid this unwanted reaction by using catalyst-free conditions, affording modafinil with a 50% yield after 5 h.

**Scheme 12.** Photochemical preparation of (**a**) sulphoraphene and (**b**) modafinil in the presence of 2-methyltetrahydrofuran.

Electrochemical oxidations also provide an attractive alternative to the more traditional chemical methods [100]. Applying this technique, Stahl et al. have recently described the aminoxyl-mediated oxidation of alcohols and aldehydes to the corresponding carboxylic acids [101]. Using this procedure, the effective synthesis of the carboxylic acid precursor **32** of levetiracetam (**33**), catalyzed by 4-acetamido-2,2,6,6-tetramethylpiperidin-1-oxyl (ACT), should be highlighted (Scheme 13). Levetiracetam is a generic drug used to treat epilepsy and is potentially beneficial for other central nervous system disorders like Alzheimer's disease and autism [102]. The use of 5 mol% ACT in an aqueous medium, applying a

potential of 0.7 V vs. Ag/AgCl and a 1.1 equivalent of NaHCO$_3$ as the electrolyte (pH 8.5) in a 40 g scale of alcohol **31**, afforded the desired compound **32** with a 91% yield and 92% *ee*.

**Scheme 13.** Electrochemical methodology for the preparation of a levetiracetam precursor (*S*)–**32**.

A few years later, Stahl et al. developed the 200 g scale procedure for the synthesis of compound (*S*)-**32** by implementing flow chemistry conditions in combination with the electrochemical protocol. Under an optimal procedure, this methodology improves the previous results in terms of yield and enantioselectivity (92%, 99% *ee*) [103].

## 4. Conclusions

Some valuable active pharmaceutical ingredients and their precursors are obtained through catalytic oxidative processes. This is especially true, for instance, in the preparation of chiral sulfoxides, compounds presenting valuable biological activities with several applications in pharmaceutical chemistry. Classical methodologies for the development of oxidative processes generally employed harsh reaction conditions and reagents (hydroperoxides and metal complexes), with the concomitant generation of wastes. Currently, several efforts are being undertaken for the development of greener solvents and catalysts to fulfil the requirements of Green Chemistry for the development of more sustainable oxidations. More specifically, the synthesis of APIs is continually being improved by the implementation of green methodologies. Thus, the development of greener solvents and catalysts and the application of techniques with low or no impact on the environment are current hot topics within the pharmaceutical industry. Most of the examples described in the last years made use of biocatalytic approaches that employ oxidative catalysts. Biocatalysis offers several advantages, as they fulfill some of the principles of Green Chemistry; not only are biocatalysts produced from renewable resources, but they are biodegradable, non-toxic, and non-hazardous, usually work under mild reaction conditions, and are active in water and biobased solvents, with a high selectivity. However, there are still multiple bottlenecks in the complete implementation of biocatalytic oxidations in the industrial synthesis of APIs. Most of the oxidative enzymes require expensive cofactors that need to be recycled, thus, making the reaction systems more complicated. Inhibition and stability issues are also serious problems for these processes, problems that have yet to be overcome in the last years despite the preparation of improved enzymes using different techniques. In conclusion, biocatalysis is a real alternative for carrying out sustainable selective oxidations in the preparation of APIs, as shown in the described preparations of prazoles, islatravir, captopril, and modafinil, among others. In addition, during the last years, other greener synthetic approaches have been analyzed for the oxidative synthesis of APIs, including the use of light or light-mediated reactions and the application of electrochemical synthesis. All of these techniques, together with the use of more sustainable solvents and reagents, have allowed chemists to perform green procedures for the preparation of Active Pharmaceutical Ingredients.

**Author Contributions:** Conceptualization, G.d.G.; methodology, P.D.G.-F., J.M.C.-C. and G.d.G.; writing—original draft preparation, P.D.G.-F., J.M.C.-C. and G.d.G.; writing—review and editing, P.D.G.-F., J.M.C.-C. and G.d.G. All authors have read and agreed to the published version of the manuscript.

**Funding:** This research received no external funding.

**Conflicts of Interest:** The authors declare no conflict of interest.

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
