# Peer review of "Green Oxidative Catalytic Processes for the Preparation of APIs and Precursors"

_catalysts, doi:10.3390/catal13030638_

Round 1
Reviewer 1 Report
This manuscript is a good compilation of the most recent advances in the synthesis of biopharmaceuticals; in particular, the last contributions of biocatalyzed processes are described.
The writing style is nice and the structure of the paper is clever. Although, the manuscript requires of several corrections before being accepted for publication, as follows:
Figure 1- The size of the text bellow the blak boxes or the all figure should be increased: Moreover, this is also an scheme and not a plot or Figure
Page 4 line 121- use the past tense as follows: “is released as wáter”
Page 4 line 126- instead of the verbe “can “ the verbe “must” is better: “…this cofator must be”
Page 4 line 131: a reference should be given at the end of this paragraph
Schemes- several ones are not refered in the text. The proper reference should be indicated for Schemes 2, 3,5,6,8 and 10
Phraseology- Some phrases are excesively complexes and could give rise to missunderstandings. Page 5 L 157-171- this long and complex phrase could be better divided in two, leaving the first phrase after ref 46. The same for phrase from L175-180 and phrase from L182 to L188
Page 6 line 208- use the past tense as follows: “was removed from…” and give the corresponding reference at the end of paragraph on Line 213
Page 6- the long phrase starting on L214 whould be better divided in two.
Page 6 L226- the excellent selectivity is attributed to the catalyst, while the complete conversión is attributed to the reaction. Also, the verbe tense on line 129 is not correct: “… which will allow…” is the appropiate tense
Page 6 L 237- Please clarify the following statements: “The main problem for these enzymes is that they usually present low activity, narrow substrate specificity and low thermostability, which makes difficult its application at industrial scale” . This is contradictory with respect to the fact that one of the limitations of biocatalyst for industrial application is their high specificity. In fact, the phrase “.drawbacks, as the (relatively) high substrate specificity of biocatalysts” is indicated on Page 3 L108.
Page 8 L296 and 297 the verb “employed” should be omited, as follows: “….The biocatalyst was immobilized..”
Page 9 L323-324- Instead of this phrase, the following order is better “..higher yields than in only buffer were achieved with several of these compounds”
Page 9 L350- “as shown in Scheme 9” rather than “scheme 8”.
Page 120 L360- a preposition is lacking: “..including the use of water or biobased..”
References
All DOIs of electronic papers should be given
Editors and pages for refs 6, 7 are lacking
Ref 13 and 62- some authors are missing
Ref 19- pages are missing
Refs 27, 30, 56, periods after each authors name are missing
Author Response
This manuscript is a good compilation of the most recent advances in the synthesis of biopharmaceuticals; in particular, the last contributions of biocatalyzed processes are described.
The writing style is nice and the structure of the paper is clever. Although, the manuscript requires of several corrections before being accepted for publication, as follows:
Figure 1- The size of the text bellow the blak boxes or the all figure should be increased: Moreover, this is also an scheme and not a plot or Figure
The size of the text bellow the boxes has been increased and some misspellings have been corrected in this Figure, now entitled Scheme 1.
Page 4 line 121- use the past tense as follows: “is released as wáter”
We think you might mean line 123 instead of 121. We changed the present tense to past tense.
Page 4 line 126- instead of the verbe “can “ the verbe “must” is better: “…this cofator must be”
We change can for must as the referee stated.
Page 4 line 131: a reference should be given at the end of this paragraph
We have included a new reference as indicated by the referee.
Schemes- several ones are not refered in the text. The proper reference should be indicated for Schemes 2, 3,5,6,8 and 10
We have refered all the schemes in the text.
Phraseology- Some phrases are excesively complexes and could give rise to missunderstandings. Page 5 L 157-171- this long and complex phrase could be better divided in two, leaving the first phrase after ref 46. The same for phrase from L175-180 and phrase from L182 to L188
We have modified the sentences indicated by the referee in order to make all them shorter and clearer for the reader.
Page 6 line 208- use the past tense as follows: “was removed from…” and give the corresponding reference at the end of paragraph on Line 213
We changed the verb tense as indicated and we have included a new reference.
Page 6- the long phrase starting on L214 whould be better divided in two.
As indicated by the referee, we have divided the sentence in two shorter ones.
Page 6 L226- the excellent selectivity is attributed to the catalyst, while the complete conversión is attributed to the reaction. Also, the verbe tense on line 129 is not correct: “… which will allow…” is the appropiate tense
We attribute the corresponding feature to the corresponding factor. We changed “which allow” by “which will allow”.
Page 6 L 237- Please clarify the following statements: “The main problem for these enzymes is that they usually present low activity, narrow substrate specificity and low thermostability, which makes difficult its application at industrial scale” . This is contradictory with respect to the fact that one of the limitations of biocatalyst for industrial application is their high specificity. In fact, the phrase “.drawbacks, as the (relatively) high substrate specificity of biocatalysts” is indicated on Page 3 L108.
As we have stated before that sometimes biocatalysts present some issues debt to their high substrate specificity (L108), we have only focused in line 237 about the low activity and stability that sometimes CYP450 present.
Page 8 L296 and 297 the verb “employed” should be omited, as follows: “….The biocatalyst was immobilized..”
We omitted the word “employed”.
Page 9 L323-324- Instead of this phrase, the following order is better “..higher yields than in only buffer were achieved with several of these compounds”
Referee is right, we have modified the sentence as suggested by him/her.
Page 9 L350- “as shown in Scheme 9” rather than “scheme 8”.
The reference in the text of Scheme 8 was changed by Scheme 9.
Page 120 L360- a preposition is lacking: “..including the use of water or biobased..”
We added the preposition “of” to the sentence.
References
All DOIs of electronic papers should be given
We have submitted DOIs of all the electronic papers.
Editors and pages for refs 6, 7 are lacking;
These two books have no editors, they have been written by the authors indicated in the references.
Ref 13 and 62- some authors are missing
Authors missing have been included.
Ref 19- pages are missing
We have included the chapter number and the pages.
Refs 27, 30, 56, periods after each authors name are missing
Periods were added where missing,
Reviewer 2 Report
The authors discuss an updated review of a series of green oxidative reactions of general interest to chemists (in particular for the synthesis of optically active sulfoxides), by biocatalytic methods and novel techniques as flow catalysis and photocatalysis.
Although the issue discussed in this review is interesting and the manuscript is generally well organized, the article presents some problems and needs an extensively rewriting before is acceptance for publication in Catalysts.
The greener chemical synthesis of ibuprofen (scheme 1), shown a lower number of steps than the classical procedure, which included palladium catalyzed oxidation reaction in water (not reported in the scheme 1).
In Scheme 2, Biocatalyzed sulfoxidations for the synthesis of AZD6738, the structure is wrong, it is not a benzene ring but pyrimidine moiety. Wrong chemical structures are also in scheme 6 and 11, please check.
Rerefences must be rechecked (such as: 43 is not complete; 51 about the synthesis of (R)-Lansoprazole don’t correspond to what is described in the text; 71 and so on…); recently was published a paper ChemistryOpen, 2022, 11, 10, e202200160 which reports tandem oxidation processes combined with nucleophilic addition reactions to the carbonyl compound generated in situ for the synthesis of secondary alcohols.
The recycle of the green solvent, or the catalyst in the papers reported was not treated in the review, only one case (scheme 8).
Author Response
The authors discuss an updated review of a series of green oxidative reactions of general interest to chemists (in particular for the synthesis of optically active sulfoxides), by biocatalytic methods and novel techniques as flow catalysis and photocatalysis.
Although the issue discussed in this review is interesting and the manuscript is generally well organized, the article presents some problems and needs an extensively rewriting before is acceptance for publication in Catalysts.
The greener chemical synthesis of ibuprofen (scheme 1), shown a lower number of steps than the classical procedure, which included palladium catalyzed oxidation reaction in water (not reported in the scheme 1).
The greener chemical synthesis included a palladium-catalyzed oxidation in water, as we show in scheme 1. Even if we do not consider the use of metal catalysts as a green procedure, in this example the reduction in the number of steps make the palladium process greener, as less isolation/purification steps were required, thus minimazing the production of wastes.
In Scheme 2, Biocatalyzed sulfoxidations for the synthesis of AZD6738, the structure is wrong, it is not a benzene ring but pyrimidine moiety. Wrong chemical structures are also in scheme 6 and 11, please check.
We apologize for these mistakes and we have corrected them in Schemes now numbered as 3, 7 and 12.
Rerefences must be rechecked (such as: 43 is not complete; 51 about the synthesis of (R)-Lansoprazole don’t correspond to what is described in the text; 71 and so on…); recently was published a paper ChemistryOpen, 2022, 11, 10, e202200160 which reports tandem oxidation processes combined with nucleophilic addition reactions to the carbonyl compound generated in situ for the synthesis of secondary alcohols.
As stated for the other referees, we have corrected all the mistakes in the references’ section. We have included the suggested reference in the manuscript. We have included the reference in the manuscript and mention this procedure as a valuable methodology for green oxidations.
The recycle of the green solvent, or the catalyst in the papers reported was not treated in the review, only one case (scheme 8).
As suggested by the referee, that would be a great point, but unfortunately, no information on biocatalyst neither solvent recycling is given on the manuscripts we have used for the revision, with the exception of the one mentioned at Scheme 8.
Reviewer 3 Report
The article submitted is a review summarizing the last advances in the processes involving catalyzed oxidation reactions for the preparation of APIs from a Green Chemistry perspective. It highlights mainly biocatalyzed reactions but also shows some protocols employing photocatalysis and flow chemistry. It has a good selection of examples and is nice to see how it goes a step beyond, showing examples of scaling-up and/or industry applicability.
I have some major and minor comments/observations:
Major:
- References numbering should be revised. I noticed in the paragraph starting in line 189 that it started saying 'One year later...' and then the reference there is [48]. This one is from 1998, and I suppose the authors wanted to cite here [49], that it is from 2019 and referred to the process they are explaining in this paragraph.
- Regarding the aforementioned issue too, sentence from line 288 to line 291 starts with 'In 2021, Romano et al....' and cite [71]. This cite is incorrect, it should be [72], and the year of this publication is 2017. Similar issue can be easily noticed on line 359: it should be [83] instead of [82].
- On Scheme 8: I suggest to write PVA-Alginate to better describe the reaction conditions, as PVA can be seen as just poly(vinylalcohol).
- On line 332, armodafinil is labaled as 24, but then on Scheme 9 the compound is labaled as (R)-24. This is redundant, as armodafinil is the R isomer. Then on line 419 modafinil is named as 24. I suggest to eliminate (24) from line 332 and add (24) next to modafinil on line 333 to fix this inconsistency.
- line 376: 'a iodine one' what does that means?
- Lines 403 to 408: The affirmation of 'During the last decades, synthetic organic photochemistry has not been considered by the chemical industry' sounds inaccurate, considering that several R&D teams from pharma and chemical industries have been focused on synthetic photochemistry during the last years together with the development of flow chemistry. I suggest to reformulate this sentences saying that 'Despite the high energy demand of most artificial light sources, the cleanness of light and effectiveness of this processes and the development of new technologies have aroused the interest of various research groups in both, academia and industry, to develop sustainable procedures employing photocatalysis. This can be supported with some references, like: https://doi.org/10.1016/j.checat.2021.12.015
https://doi.org/10.1021/acs.oprd.8b00213
https://doi.org/10.1016/j.trechm.2019.09.003
- Paragraph on line 431: this same protocol was later optimized by Stahl group too, for large scale production n a continuous flow protocol, allowing to prepare 200 g of product. I believe this process is worth to be mentioned in this review. This is the reference: https://doi.org/10.1021/acs.oprd.1c00036
- On line 557: Reference 43 just says 'Prazoles review'. Please revise this reference.
Minor:
There are some expression from English that should be revised and several typos along the text that should be corrected. I assumed that more proofreading will be done on the manuscript but anyway here are some of them I noticed:
line 44: 'are ensured' is repeated from line 43 and have to be eliminated.
line 47: 'provide to reflect about' expression should be revised. It can be maybe changed for 'provide a reflection on'.
Figure 1: On principle 3, it should be hazardous instead of hazardious.
Scheme 1: on H+/H2O of step 3, the oxygen atom of water is in a second line.
line 158: there is a ] after Esomeprazole.
line 321: was due instead of was debt.
line 345: it should be 23 instead of 21 (and it should be written bold).
line 350: it is Scheme 9 instead of Scheme 8.
line 369: I suggest tu use the IUPAC name or the most common name and its acronym, 1,1,1,3,3,3-hexafluoropropan-2-ol, hexafluoroisopropanol, HFIP.
line 383: it should be 26 instead of 24.
line 393: it should be 'of' instead of 'or'.
line 433: it is aminoxyl instead of aminoxil.
######
I consider this article has a good fit with the journal scope and it is acceptable for publication after suggested revisions are met.
Author Response
The article submitted is a review summarizing the last advances in the processes involving catalyzed oxidation reactions for the preparation of APIs from a Green Chemistry perspective. It highlights mainly biocatalyzed reactions but also shows some protocols employing photocatalysis and flow chemistry. It has a good selection of examples and is nice to see how it goes a step beyond, showing examples of scaling-up and/or industry applicability. I have some major and minor comments/observations:
Major:
- References numbering should be revised. I noticed in the paragraph starting in line 189 that it started saying 'One year later...' and then the reference there is [48]. This one is from 1998, and I suppose the authors wanted to cite here [49], that it is from 2019 and referred to the process they are explaining in this paragraph.
Referee is right, we have corrected this mistake.
- Regarding the aforementioned issue too, sentence from line 288 to line 291 starts with 'In 2021, Romano et al....' and cite [71]. This cite is incorrect, it should be [72], and the year of this publication is 2017. Similar issue can be easily noticed on line 359: it should be [83] instead of [82].
We have corrected these mistakes in the references numbering.
- On Scheme 8: I suggest to write PVA-Alginate to better describe the reaction conditions, as PVA can be seen as just poly(vinylalcohol).
We have modified this, indicating PVA-alginate in the scheme.
- On line 332, armodafinil is labaled as 24, but then on Scheme 9 the compound is labaled as (R)-24. This is redundant, as armodafinil is the R isomer. Then on line 419 modafinil is named as 24. I suggest to eliminate (24) from line 332 and add (24) next to modafinil on line 333 to fix this inconsistency.
In order to avoid this inconsistency, we have labelled 24 as modafinil, moving the number from line 332 to 333. We have included (R)-24 to refer to armodafinil.
- line 376: 'a iodine one' what does that means?
That was a mistake, we wanted to mean “iodine”. We have corrected it.
- Lines 403 to 408: The affirmation of 'During the last decades, synthetic organic photochemistry has not been considered by the chemical industry' sounds inaccurate, considering that several R&D teams from pharma and chemical industries have been focused on synthetic photochemistry during the last years together with the development of flow chemistry. I suggest to reformulate this sentences saying that 'Despite the high energy demand of most artificial light sources, the cleanness of light and effectiveness of this processes and the development of new technologies have aroused the interest of various research groups in both, academia and industry, to develop sustainable procedures employing photocatalysis. This can be supported with some references, like: https://doi.org/10.1016/j.checat.2021.12.015, https://doi.org/10.1021/acs.oprd.8b00213, https://doi.org/10.1016/j.trechm.2019.09.003.
We have modified the sentence as the referee has indicated. We have also included the suggested references.
- Paragraph on line 431: this same protocol was later optimized by Stahl group too, for large scale production n a continuous flow protocol, allowing to prepare 200 g of product. I believe this process is worth to be mentioned in this review. This is the reference: https://doi.org/10.1021/acs.oprd.1c00036
We accept the suggestion and include the mentioned example.
- On line 557: Reference 43 just says 'Prazoles review'. Please revise this reference.
There was a mistake and we have removed it as we have included other references about prazoles in the text.
Minor:
There are some expression from English that should be revised and several typos along the text that should be corrected. I assumed that more proofreading will be done on the manuscript but anyway here are some of them I noticed:
line 44: 'are ensured' is repeated from line 43 and have to be eliminated.
We eliminated the suggested expression
line 47: 'provide to reflect about' expression should be revised. It can be maybe changed for 'provide a reflection on'.
We have changed the suggested expression as the referee stated.
Figure 1: On principle 3, it should be hazardous instead of hazardious.
This misspelling has been corrected.
Scheme 1: on H+/H2O of step 3, the oxygen atom of water is in a second line.
We corrected the mistake on the scheme.
line 158: there is a ] after Esomeprazole.
The symbol was deleted.
line 321: was due instead of was debt.
We changed “debt” for “due”
line 345: it should be 23 instead of 21 (and it should be written bold).
We corrected the number of the compounds
line 350: it is Scheme 9 instead of Scheme 8.
We changed the number of the scheme
line 369: I suggest tu use the IUPAC name or the most common name and its acronym, 1,1,1,3,3,3-hexafluoropropan-2-ol, hexafluoroisopropanol, HFIP.
We changed the name that we have provided for the IUPAC name.
line 383: it should be 26 instead of 24.
We have corrected the compound number.
line 393: it should be 'of' instead of 'or'.
We have written ‘of’ instead of ‘or’
line 433: it is aminoxyl instead of aminoxil.
We have corrected this misspelling.
######
I consider this article has a good fit with the journal scope and it is acceptable for publication after suggested revisions are met.
Round 2
Reviewer 2 Report
The authors appropriately reviewed their manuscript according to the reviewers' comments.